# Adaptive Trajectories' Constant False Alarm Rate Mirror Filters and Elevation Angle Evaluation for Multiple-Input Multiple-Output Radar-Based Hand Gesture Recognition

**Tzu-Jung Tseng and Jian-Jiun Ding *** 

Graduate Institute of Communication Engineering, National Taiwan University, Taipei City 10617, Taiwan; d09942007@ntu.edu.tw
* Correspondence: jjding@ntu.edu.tw; Tel.: +886-2-33669652

**Abstract:** Gesture recognition technology has been quickly developed in the field of human–computer interaction. The multiple-input multiple-output (MIMO) radar is popular in gesture recognition because of its notable spatial resolution. This work proposes a MIMO radar-based hand gesture recognition algorithm with low complexity. We leverage low-complexity adaptive signal processing to extract trajectory information and minimize noise to create a system that can be applied in real-world applications with small training datasets. First, a spectrum analysis is utilized on range-Doppler maps (RDMs), and a cell-averaging constant false alarm rate (CA-CFAR) with mirror filters is applied to improve the robustness of noise. Then, the features related to the distance, speed, direction, and elevation angle of the moving object are determined using the proposed adaptive signal analysis techniques. For classification, the random forest algorithm is implemented. The proposed system can precisely distinguish and identify eight gestures, including waving, moving to the left or right, patting, pushing, pulling, and rotating clockwise or anti-clockwise, with an accuracy of 95%. Experiments demonstrate the capability of the proposed hand gesture recognition system to classify different movements precisely.

**Keywords:** MIMO radar; hand gesture recognition; adaptive signal processing algorithm; random forest; mirror filtering; elevation angle estimation





## 1. Introduction

In recent years, the radar sensor has become a popular non-contact technology known for its precision, independence from visible light, and robust performance in diverse environments. This technology is applicable in object detection [1], healthcare [2,3], and automobile industries [4,5]. Several researchers are exploring various radar models for hand gesture recognition, including the continuous wave radar (CW), the frequency-modulated continuous wave radar (FMCW), and the multiple-input multiple-output (MIMO) radar [6]. In particular, MIMO radars achieve higher resolution by leveraging advanced processing and spatial diversity, which broadens the field of perception with their multi-antenna performance.

The MIMO technology simultaneously uses multiple antennas and channels to achieve spatial diversity, which requires heavy computation to process received information. Several learning-based hand gesture recognition algorithms have been developed recently [7–9]. These approaches involve large labeled training datasets and sophisticated computational architecture to improve precision. In addition, they usually need advanced computing hardware, such as graphic processing units (GPUs), which limits their feasibility in resource-constrained scenarios. Therefore, we propose an economical classification technique for MIMO radar-based hand gesture recognition.

This paper proposes a high-accuracy hand gesture recognition system with low complexity, which applies a range-Doppler map (RDM) to generate range parameters adaptively

and uses the elevation angle information. The proposed algorithm applies adaptive signal processing techniques for a trajectory analysis, uses smoothing and mirror filters to reduce the noise effect, and adopts the elevation angle information to improve accuracy. The random forest classifier is implemented to achieve good performance even when the training data size is limited. With the proposed algorithm, eight distinct hand gestures, including waving, moving to the left or right, patting, pushing, pulling, and rotating clockwise or anti-clockwise, can be recognized accurately. Experiments show that, with the proposed adaptive signal processing techniques and the random forest classifier, an accuracy of 95% can be achieved.

Hence, this article aims to develop a low-complexity algorithm based on a single MIMO radar for hand gesture recognition by leveraging information on the Doppler velocity and the spatial location obtained from detected objects. The main contributions of our solution are as follows:

1. This study uses adaptive signal processing techniques, including CA-CFAR, MUSIC, and an elevation angle analysis, to extract the characteristics of the temporal trajectory of moving objects instead of using range-Doppler or range-angle images.

2. The classification algorithm is based on the random forest method, which reduces computing costs compared to the methods based on deep learning.

3. Cooperating with a single-chip MIMO radar, our algorithm creates a low-complexity radar gesture recognition system suitable for healthcare and automatic driving applications.

The structure of this article is as follows: The background and related work are reviewed in Section 2. Section 3 provides a detailed illustration of each part of the proposed low-complexity hand gesture algorithm. Several experiments are conducted to evaluate the performance of the algorithm. A conclusion is made in Section 5.

## 2. Preliminary Work

There are several existing radar-based gesture recognition algorithms. They are based on different hardware architectures, including the continuous wave (CW) radar [10–12], the frequency-modulated continuous wave (FMCW) radar [7,8,13–20], and the multiple-input multiple-output (MIMO) radar [8,21,22]. The CW radar is simple to design and has excellent Doppler sensitivity, but it lacks the resolution of range. The FMCW radar can measure range and velocity, but it is significantly impacted by noise. The MIMO radar increases target localization by enhancing the angular resolution and reducing the effect of clutter.

Several studies have been conducted using the CW radar to recognize hand gestures. Skaria et al. [10] used low-cost radar chips and advanced machine learning techniques. They used two-antenna Doppler radar to record the Doppler signatures of hand gestures and then trained a neural network. In [11], Wang et al. introduced an innovative approach to recognizing dynamic hand gestures by analyzing micro-Doppler radar signals using Hidden Gauss–Markov Models.

In 2022, Leu et al. [15] applied the RDM in a gesture detection system. They used the NN as the feature extractor. Zhao et al. [16] proposed a gesture detection system using the FMCW radar. They implemented Blender 4.0 to generate different hand gestures and trajectories. They employed a two-dimensional Fast Fourier Transform (FFT) and synthetic feature extraction.

With the adaptability of the spatial domain, there is an increase in the amount of research in MIMO human–computer interaction (HCI). Lee et al. [8] used two different radar and Lei et al. [21] used the IWR1443 MIMO radar to generate the RDM and the range-azimuth map (RAM) in ten gestures, leveraging the data fusion technology coupled with the 3DCNN-LSTM network architecture. Zheng et al. [22] developed a transformer-based network with range-Doppler, range-angle, and range-elevation data frames. Alirezazad and Maurer [23] explored a 77-GHz MIMO FMCW radar and deep learning for the automated contactless recognition of human hand gestures, which involved training 2D CNNGRU architecture by feeding range-Doppler and range-angle images. We compared the above

research with different hardware architectures and related algorithms in Table 1. These studies use deep learning-based approaches with massive labeled training datasets and powerful computing architectures for accuracy, which restricts their adaptability in resource-constrained applications. In contrast, in this work, we plan to propose an inexpensive method for MIMO radar-based hand gesture identification.

**Table 1.** Gesture recognition comparison for MIMO radar-based methods.

| Methods | MIMO Device | Frequency | Antennas | Gestures | Algorithm |
| --- | --- | --- | --- | --- | --- |
| [8] | IWR6843-AOP + MMWCAS-DSP/RF-EVM | 60–64 GHz+ | 3T4R + 12T16R | 8 | LSTM |
| [21] | IWR1443 | 76–81 GHz | 3T4R | 10 | 3D-CNN + LSTM |
| [22] | IWR6843-AOP | 60–64 GHz | 3T4R | 8 | RGTNet |
| [23] | Radarbook2 | 77 GHz | 2T16R | 10 | dual-stream 2D CNN-GRU |

There are many classical classification algorithms in machine learning, including neural networks (NNs), support vector machines (SVMs), and random forests. The SVM was adopted for hand gesture recognition in [16]. Based on the random forest, study [9] aimed to determine five different emotions by calculating the relative positions of joints in the arms captured from the Kinect sensor and achieved an accuracy of 86.8%. Hao et al. [20] proposed a method with millimeter-wave near-field synthetic-aperture radar (SAR) imaging for wireless static gesture recognition. They used a histogram of oriented gradients for feature extraction and applied a principal component analysis for dimensionality reduction. Tsang et al. [24] proposed an algorithm using different classifiers in the identification system, such as logistic regression, the random forest, and the SVM. The detailed abbreviations and definitions used in this paper are listed in Table 2.

**Table 2.** List of abbreviations and acronyms used in this paper.

| Abbreviation | Definition | Abbreviation | Definition |
| --- | --- | --- | --- |
| ADC | Analog-to-Digital Converter | KNN | k-Nearest Neighbor Algorithm |
| CA-CFAR | Cell-Averaging Constant False Alarm Rate | MIMO | Multiple-Input Multiple-Output |
| CFAR | Constant False Alarm Rate | MUSIC | Multiple Signal Classification |
| CW | Continuous Wave | NN | Neural Network |
| DoA | Direction of Arrival | RDM | Range Doppler Map |
| FFT | Fast Fourier Transform | RAM | Range Angle Map |
| FMCW | Frequency Modulated Continuous Wave | SAR | Synthetic-Aperture Radar |
| HCI | Human–Computer Interaction | SVM | Support Vector Machine |

## 3. Materials and Methods

This study presents a specialized MIMO radar sensor to identify and analyze hand gestures. In Figure 1, the diagram of the proposed architecture is depicted. The radar's 12 virtual antennas gather the trajectories of hand gestures. The classification of hand gestures is systematically partitioned into four distinct stages. The data are first processed using a two-dimensional Fast Fourier Transform (FFT) to compute range and velocity. The next stage applies a refined constant false alarm rate mechanism, which enhances the ability to locate the spatial coordinates of the palm. The multiple signal classification (MUSIC) technique determines the angle values. After obtaining these parameters, feature

extraction is performed, and the obtained features are fed into the random forest method for classification.

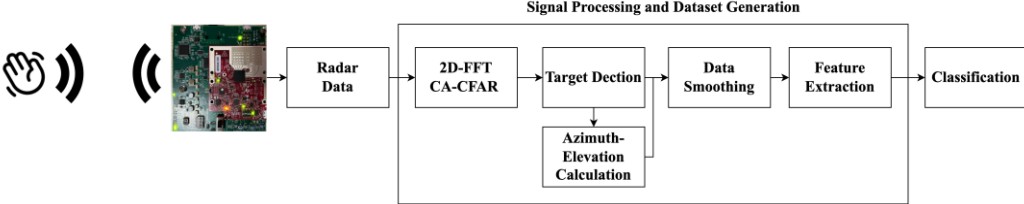

**Figure 1.** The overview of the proposed hand gesture recognition.

### 3.1. Range and Velocity Analysis

The combination of the MIMO radar with the FMCW technology takes the advantages of the MIMO system in spatial diversity and the FMCW system in remarkable frequency modulation performance. The fusion leads to a highly adaptive, precise, and efficient radar system. Its detail is provided as follows: First, the radar transmits the chirp-modulated waveform through a transmitting antenna. It can be represented as

$$S_T = A_T \cos\left(2\pi f_c t + \pi \frac{B}{T_c} t^2\right) = A_T \cos(\phi_T(t)), \tag{1}$$

where $f_c$ is the carrier frequency, $B$ is the sweep bandwidth of the chirp, and $T_c$ is the chirp duration. On the other hand, the received signal is the signal echoed from a target, which is a scaled and delayed version of the transmitted signal.

$$S_R = \alpha S_T(t - t_d) = \alpha A_T \cos\left(2\pi f_c(t - t_d) + \pi \frac{B}{T_c}(t - t_d)^2\right), \tag{2}$$

where $\alpha$ is a scaled factor and $t_d$ is the delayed time. After that, the received signal is mixed with the transmitted signal to create an intermediate frequency signal:

$$S_{IF} = S_T \times S_R = \frac{\alpha A_T}{2}[\cos(\phi_T(t - t_d) - \phi_T(t)) + \cos(\phi_T(t - t_d) + \phi_T(t))], \tag{3}$$

where the beat frequency after the receiving mixer is represented as $f_b = B t_d / T_c$. For static objects, the beat frequency is proportional to the distance, which is fulfilled by taking the Fast Fourier Transform (FFT) of the received IF signal. However, the velocity is determined for moving objects using phase change across multiple chirps. The phase and the frequency of the received signal change with the velocity of the moving object. A second FFT is then applied across these chirps to extract the information about phase variation and velocities. This process yields a comprehensive 2D range-Doppler map, providing valuable insights into the spatial distribution and velocity characteristics of detected objects.

Besides a traditional radar system with one single antenna transmitter, MIMO radar uses multiple antennas for transmission, which can create multidimensional arrays for spatial diversity. In our scenario, the physical antenna configurations with three transmitting antennas and four receivers can extend to 12 virtual antennas. The raw radar data are represented by a frame with three dimensions, ADC sampling, chirps, and antennas, as shown in Figure 2. In the ADC sampling dimension, ranges are extracted using the FFT in a process known as the fast-time FFT or the range-FFT. Furthermore, velocity information can be obtained by employing the FFT, which is referred to as the slow-time FFT or the Doppler-FFT from chirp dimensions. As depicted in Figure 2, the range-Doppler map (RDM) is generated using the two-dimensional FFT, providing an accurate target range and velocity data. The hand motion can be depicted by analyzing the variations in range and velocity recorded in the frames of the RDM. To determine the exact palm position, interference from noise must be minimized.

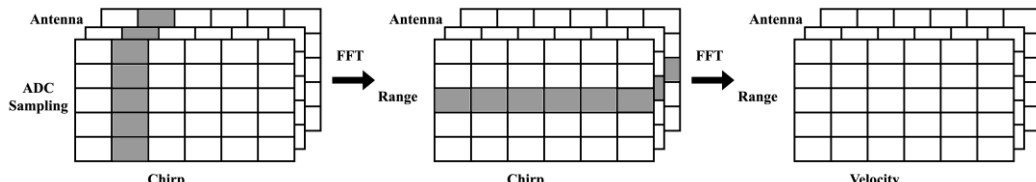

**Figure 2.** Radar signal pre-processing for the RDM.

*3.2. CFAR and Target Detection*

When detecting the target, the background noise and other non-target signals will increase the difficulty of detecting the target. Therefore, a constant false alarm rate (CFAR) is applied to achieve reliable and consistent detection accuracy. The CFAR technique is based on statistical and signal processing concepts. It utilizes local information and adjusts the detection threshold to ensure a consistent false alarm rate across different conditions. The proposed algorithm uses a cell averaging (CA)-CFAR method, which involves identifying a set of reference cells surrounding the target cell and calculating its average power. An object is classified as a target if its average surpasses a specified threshold. Despite the noise-filtering capabilities of the CA-CFAR technology, human hands, arms, and body movements can still lead to inaccurate predictions. In order to obtain precise hand gesture data from the radar, the proposed system resolves the following issues:

1. Zero-Velocity Disturbances.

The presence of zero-velocity disturbances during an RDM analysis may complicate the interpretation of data. One common source of zero-velocity disturbances is the unwanted reflection from the environment, which remains after the CFAR. The clutter may be mistakenly viewed as target movement. Thus, the proposed method incorporates an additional two-dimensional filter to differentiate between stationary and moving objects and enhance the accuracies of target identification and trajectory tracking. In order to obtain precise information from the radar and reduce these disturbances, incorporating an additional filter is a beneficial approach for radar system calibration.

2. Peaks caused by other body parts.

After determining the CA-CFAR threshold using a fixed multiplier of the penetration average power, we need to filter the motion of other body parts to detect hand motion. Compared to other objects, the palm will move at the fastest speed and be closest to the radar. Therefore, we will determine the position of the hand based on these two conditions. It will raise the ability of the radar system to detect complex trajectories.

3. The target velocity wrapping.

The determination of the target velocity is challenging when it surpasses the predetermined velocity range and returns to the opposite end of the velocity spectrum by wrapping or folding. In order to tackle this problem, we utilize the mirroring technique that is commonly used in the field of image processing. More precisely, we expand the data by replicating the initial three and final three columns, providing a continuous variation of the speed. Assume that the RDM is $X(f_d, f_r, t)$. Then, the result after applying CA-CFAR is

$$X_{CA-CFAR}(i,j) = \begin{cases} 1, & \text{if} |X(f_d, f_r, t)| > \gamma * local\_average(i,j) \\ 0, & \text{otherwise} \end{cases}, \qquad (4)$$

where $\gamma$ is the threshold factor and $local\_average(i,j)$ is the average reflectivity within a specific region. After that, we mirror the first and last three rows of the $X_{CA-CFAR}(i,j)$, which can be represented as

$$X_{CA-CFAR}(i,j) = \begin{cases} X_{CA-CFAR}(n-i+1,j), & \text{if } i \leq 3 \text{ } or \text{ } i > n-3 \\ X_{CA-CFAR}(i,j), & \text{otherwise} \end{cases}, \qquad (5)$$

where $n$ is the number of rows in the matrix.

By implementing this mirrored technique, we successfully address the issue related to the target velocity beyond the specified range. This technique ensures the precise capture of the goal distance and velocity of trajectories while preserving the smoothness of the speed.

### 3.3. Angle of Arrival

In addition to determining the range and the velocity, the MIMO radar can also determine the angle between the radar and the target. By employing the Direction of Arrival (DoA), one can precisely identify the angles at which different objects are positioned. If there exists $N_{TX}$ transceivers and $M_{RX}$ receivers, and $L$ and $\lambda$ represent the distance between antennas and the wavelength, respectively, then the angle resolution $\theta_{res}$ can be determined from

$$\theta_{res} = \frac{\lambda}{N_{TX}M_{RX} \times Lcos(\theta)} \frac{180°}{\pi}, \tag{6}$$

Equation (6) shows that the hardware structure still restricts the angle resolution. Therefore, we employ a Direction of Arrival (DoA) estimation approach to enhance the angle resolution.

The DoA methods help calculate the degrees to which signals from reflected targets are received. By analyzing the DoA, one can determine the angles of the targets. DoA estimation can be classified into four main categories: beamforming, maximum likelihood, subspace-based methods, and compressive sensing. This paper utilizes the well-known MUSIC method [25], a subspace-based method, by using the eigen-structure of the data covariance matrix. Assume that the matrix of the received signals by one receiving antenna is x($t$) of size $M \times 1$. Also, suppose that transmission signals are s($t$) $\in \mathbb{C}^{d \times 1}$ and the noises are n($t$) $\in \mathbb{C}^{M \times 1}$. Then, the received signals' matrix at time $t$ can be written as

$$x(t) = A(\theta)s(t) + n(t), \tag{7}$$

where $A(\theta) \in \mathbb{C}^{M \times d}$ is the steering matrix.

Since signals and noises are uncorrelated, the covariance matrix of x($t$) is

$$R_{xx} = E\left\{x(t)x^H(t)\right\} = A(\theta)R_{ss}A^H(\theta) + \sigma_n^2 I_M, \tag{8}$$

where $R_{ss}$ is the signal correlation matrix, $\sigma_n^2$ is the noise variance, and $I_M$ is the $M \times M$ identity matrix. Next, suppose that $d \leq M$. Let $\Sigma_s = diag(\lambda_1, \lambda_2, \ldots, \lambda_d)$ and $\Sigma_n = diag(\lambda_{d+1}, \lambda_{d+2}, \ldots, \lambda_M)$ be the corresponding eigenvalues of the signal and noise eigenvectors, respectively. Based on the assumption in MUSIC, the steering vector of the signal $a^H(\theta)$ is orthogonal to the subspace $U_N$ of noise, which can be written as

$$a(\theta)^H U_N = 0, \tag{9}$$

where $a(\theta)$ is a column of matrix $A(\theta)$ with $A(\theta) = [a(\theta_1), a(\theta_2), \ldots, a(\theta_M)]$. Then, Equation (8) can be represented as

$$R_{xx} = U_s\Sigma_s U_s^H + U_n\Sigma_n U_n^H = U\Sigma_x U^H, \tag{10}$$

with $U_x = [U_s, U_n]$ and $\Sigma_x = diag(\lambda_1, \lambda_2, \ldots, \lambda_M)$. Then, the MUSIC spectrum is defined as

$$P_{MUSIC}(\theta) = \frac{1}{a^H(\theta)U_n U_n^H a(\theta)}. \tag{11}$$

Meanwhile, by increasing the number of snapshots used for spectrum estimation, the ability of MUSIC to separate multiple targets can be further improved.

### 3.4. Elevation Calculation

To determine elevation angles precisely and reliably, an innovative method suitable for the IWR6843ISK radar system is proposed. It is designed with two antennas positioned at the same azimuth angle. As indicated with Equation (6), this configuration results in a resolution angle of 45 degrees.

The structure of the radar system provides a collection of four elevation angles obtained with antenna information. The possible location of the target is either on the upper side or the opposite side of the radar. Moreover, this additional layer improves the combination of data sources, thereby reducing the impact of the hardware limitation. In order to improve the accuracy, a voting mechanism is implemented. The voting system will determine the most possible elevation angle. This mechanism enhances the dependability of elevation angle estimation, resulting in a more precise depiction of data. Although limited by hardware, the proposed elevation angle determination algorithm is able to overcome the technological challenge and maintains the precision of the estimation result.

### 3.5. Data Smoothing and Feature Extraction

The hand position in each RDM was recorded and then used to determine the angle of hand motion. The MUSIC method was used to calculate the angle using the hand position data collected from eight antennas in the RDM. The elevation was determined using the method mentioned above. As a result, we obtained the measurements for the range, the velocity, the azimuth, and the elevation angle of the dynamic hand movements in each frame.

Moreover, a data smoothing approach was applied due to the fluctuation of the detected range, velocity, and angle values. The moving average technique with window size three was utilized to enhance the precision of the data and reduce variations. This technique helps improve the robustness to noise and enables a more organized interpretation of hand movements. Then, the detected range, velocity, and angle values are stored in a $4 \times n$ matrix $T$, where each column is represented as

$$T_{r,v,a,e} = [R_i, V_i, A_i, E_i]^T, \text{where } i = 1, 2, \ldots, n. \tag{12}$$

Then, the moving average of $T$ is computed:

$$\hat{T}_{r,v,a,e}(i) = \frac{1}{3} \sum_{j=1}^{3} T_{r,v,a,e}[k, i-j], \text{where } k = 1, 2, 3, 4. \tag{13}$$

Next, we created a set of features to improve the classification of different movements using different parts of the recorded data.

During feature extraction, we employ two approaches to segment the data. One approach involves analyzing the data based on the total duration of object movement, and the other entails dividing the data into eight equal segments for an individual analysis. We extract features from the collected data in our study, which includes capturing the maximum and minimum values of the velocity and azimuth angles, determining their ranges and distributions, and assessing the differences between the initial and final values of each parameter. We also determine the differences between these parameters and roughly estimate elevation angles. By incorporating these subtle features, we aim to provide a comprehensive representation of the dynamic aspects of hand trajectories, which will help improve the accuracy of the later classification process.

### 3.6. Classification

In this work, the random forest algorithm has been used as the classification method. The random forest is a machine learning ensemble technique that explicitly tackles these issues. During the training process, a designed number of decision trees are created, and each is built by randomly selecting a subset of features. Moreover, a voting method is

adopted to predict these trees. The variation of feature and sample selection improves the model's flexibility, reduces the likelihood of overfitting, and enhances generalization.

## 4. Results and Discussion

In this section, a comparative analysis of the proposed algorithm with three other popular machine learning approaches based on our MIMO radar system is conducted. MATLAB2022a is used to simulate all methods, and all experiments are performed on identical datasets.

### 4.1. Radar System Configuration

The Texas Instruments IWR6843 mm wave (mmWave) radar board and the Texas Instruments DCA1000 data acquisition adaptor were utilized [26,27]. The radar board was equipped with four transmitters and three receivers that function within the frequency range of 60–64 GHz. This setup enabled outstanding range and velocity resolution. To create a virtual array for measuring the azimuth angle and the elevation angle, a setup with three transmitters and four receivers was utilized, resulting in 12 antennas. Signals produced using a synthesizer were distributed via the three transmitters. Meanwhile, four receivers captured the signals after they were reflected from the target.

### 4.2. Building Gesture Dataset

The dataset was collected by five people and consisted of eight dynamic movements, each accompanied by corresponding RDM images from 12 receiving channels. The movements and their visual representations are shown in Figure 3 and explained as follows:

- Wave (W)—The palm waves for two cycles.
- Left (L)—The palm swipes from left to right.
- Right (R)—The palm swipes from right to left.
- Pat (P)—The palm moves back and forth rapidly for two cycles.
- Push (PS)—The palm pushes forward.
- Pull (PL)—The palm pulls away.
- Clockwise (C)—The palm draws a clockwise circle in the air.
- Anti-Clockwise (AC)—The palm draws an anti-clockwise circle in the air.

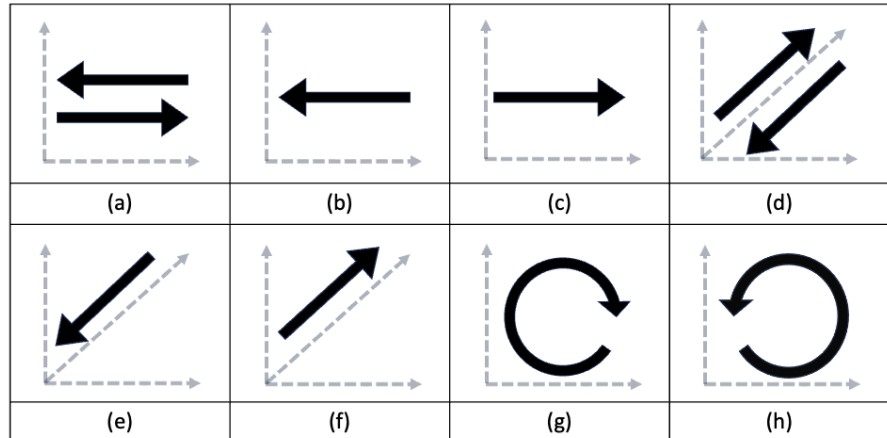

**Figure 3.** The gesture considered in experiment. (**a**) Wave; (**b**) left; (**c**) right; (**d**) pat; (**e**) push; (**f**) pull; (**g**) clockwise; (**h**) anti-clockwise.

### 4.3. Radar Image Examples

The dynamic movements are recorded and analyzed to identify specific features such as the range, the velocity, and the angle. The palm is the region of motion, and its coordinates are determined using the strongest signal in each frame. Figure 4 illustrates the pattern of palm movement throughout time. It characterizes each gesture in the RDM.

During the push (PS) action, the target moves upwards toward the top of the RDM, whereas during the pull (PL) action, it travels away from the RDM. On the right side of Figure 4, how distance and velocity vary with time is depicted. These variations prominently reveal the motion patterns to be recognized. For instance, when performing a wave (W) motion, there is a periodic variation in speed; when executing a pat (P) gesture, there is a periodic change in distance and velocity. In addition, 'push' and 'pull' exhibit distinctive distance and velocity variation characteristics. However, additional factors like the angle should also be considered to distinguish other motions.

In the proposed algorithm, the angle value is added as a parameter to differentiate between various movements. Figure 5 illustrates the distributions of different hand movements in the range-angle domain. It is also noticeable that both left and right movements exhibit lateral displacement, whereas 'pat', 'push', and 'pull' are characterized by vertical motion. However, distinguishing between clockwise and counterclockwise movements is more complex solely based on these graphs. Consequently, we turn to analyze these eight gestures through additional defined parameters.

*4.4. Gesture Recognition Performance Analysis*

The experimental hand gesture data collected by us consist of the inputs from five separate right-handed persons. All followed the instructions to perform and repeated 25 iterations of eight motions. It resulted in a total of 125 instances for each gesture. For data collection, 100 frames were captured for each gesture, with participants positioned at a distance of one meter from the radar sensor.

Our project focuses on confronting recognition challenges in a situation with limited data and with the requirement of a quick analysis. Because modern deep learning-based techniques rely on large datasets and a large computing capacity, our comparative study employs conventional machine learning methods and emphasizes the efficiency and usability of the related algorithms, including the SVM, the KNN, and the NN.

The random forest algorithm is utilized for the classification using a five-fold cross-validation method, adopting an 80–20% split between training and validation data. The adopted random forest architecture is configured with 20 trees in the ensemble. The training dataset consists of 100 sequences, while the testing dataset has 25 sequences. The confusion matrix that resulted from the analysis of the suggested approach is depicted in Figure 6. The random forest method resulted in an overall accuracy rate of 95%. The left (L) gesture, in particular, demonstrated a significantly high accuracy of approximately 99%, making it distinctly recognizable. Tables 3 and 4 present a comparative analysis using the SVM with the one-vs.-one strategy, the K-Nearest Neighbor (KNN) algorithms with *k* set to five, and the NN methods based on the feedforward technique with 50 epochs.

**Table 3.** Recognition Accuracy comparison for each method.

| Method | W | L | R | P | PS | PL | C | AC |
|--------|----|----|----|----|----|----|----|----|
| SVM | 99 | 98 | 91 | 99 | 95 | 91 | 91 | 90 |
| KNN | 90 | 98 | 92 | 95 | 94 | 88 | 81 | 92 |
| NN | 96 | 88 | 88 | 92 | 96 | 88 | 80 | 92 |
| Proposed | 94 | 99 | 94 | 97 | 95 | 92 | 93 | 92 |

**Table 4.** Recognition Accuracy comparison for each method with time.

| Method | Accuracy | Training Time | Testing Time | Energy Consumption after 450 Iterations |
|--------|----------|---------------|--------------|------------------------------------------|
| SVM | 95% | 0.17 s | 0.013 s | 2.10 Wh |
| KNN | 91% | 0.03 s | 0.005 s | 0.42 Wh |
| NN | 90% | 2.76 s | 0.021 s | 2.52 Wh |
| Proposed | 95% | 0.07 s | 0.012 s | 2.10 Wh |

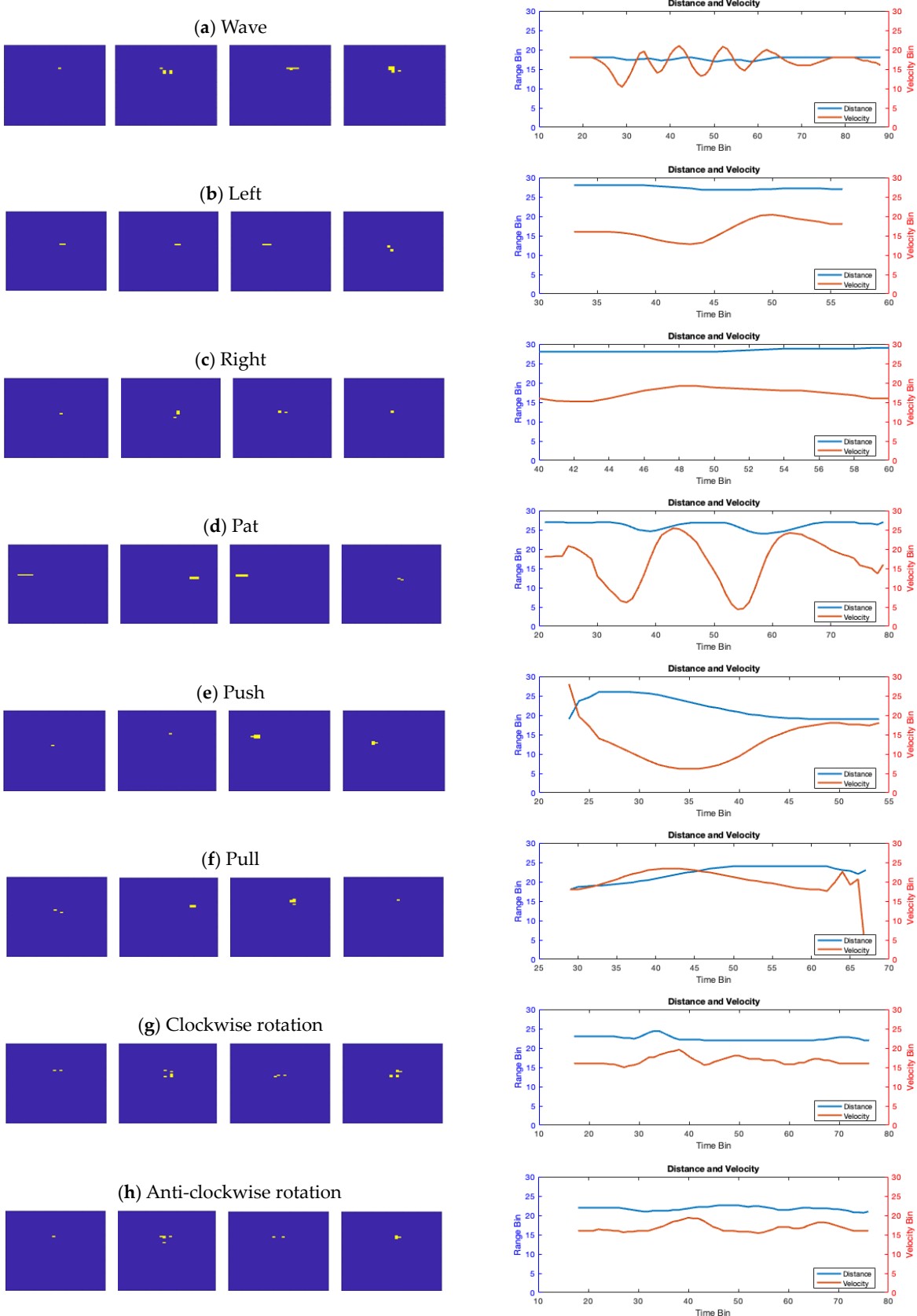

**Figure 4.** The range-Doppler map with gestures through frames. (**a**) Wave; (**b**) left; (**c**) right; (**d**) pat; (**e**) push; (**f**) pull; (**g**) clockwise; (**h**) anti-clockwise.

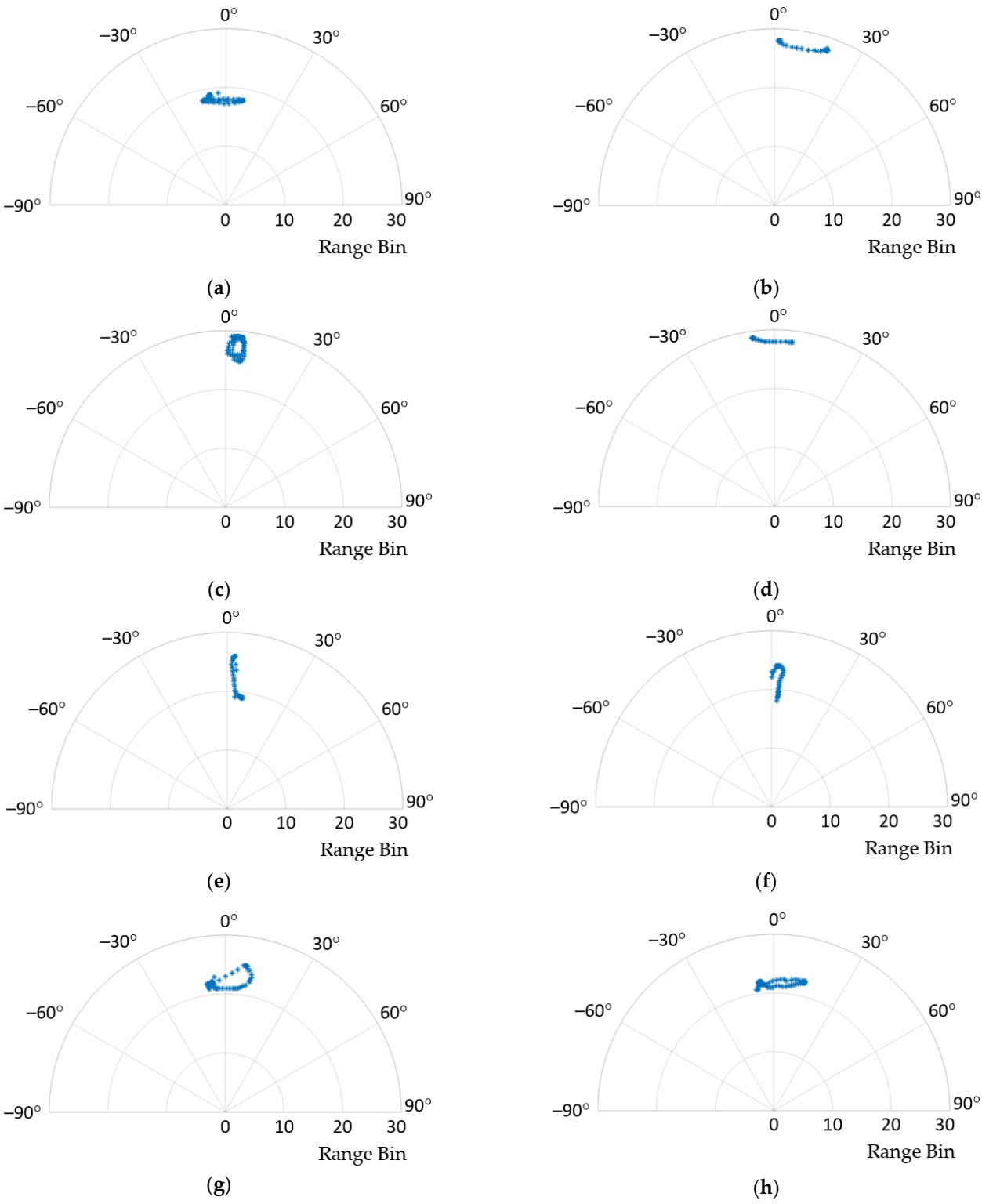

**Figure 5.** The range-angle in polar plot with different gestures. (**a**) Wave; (**b**) left; (**c**) right; (**d**) pat; (**e**) push; (**f**) pull; (**g**) clockwise; (**h**) anti-clockwise.

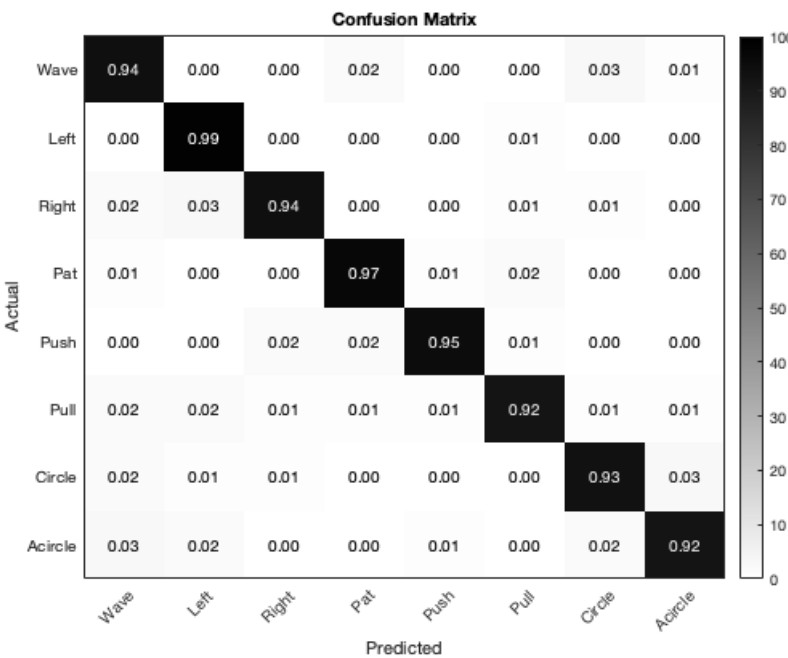

**Figure 6.** The confusion map using the proposed algorithm.

The proposed method is based on the IWR6843ISK radar model. If we use other radar devices, the dimension of the received data is different, which will affect the detection ability and the resolution of the parameters, and even impact the overall SNR of the radar. The proposed signal processing procedure is still applicable after proper calibration and modification for different devices. Please note that accuracy might be influenced by varying resolutions and data volumes.

Considering the time requirement of the four approaches, in the training phase, the SVM undertakes a complicated procedure to choose the most proper hyperplane that effectively distinguishes various classes in the feature space, which results in a slower training speed. The NN requires large amounts of data to train a classification model with many parameters, and insufficient data may lead to overfitting. Generally, SVMs and random forests can often perform well with smaller datasets. Moreover, the last column in Table 4 shows the energy consumption when performing each of the algorithms 450 times on a 12th Gen i5-1235U laptop with a 42 Wh battery. The energy consumption was computed based on the battery capacity and the percentage of the remaining energy. The results show that the proposed method requires a total of $2.1/450 = 0.00467$ Wh. Its computation time is only 0.012 s.

In summary, the proposed approach can achieve the highest accuracy with less training and testing time and less power consumption, highlighting its effectiveness in precisely identifying hand movements. In Figure 6, the confusion matrix shows the robustness of the proposed gesture classification algorithm, with each gesture reaching over 92% accuracy. The average accuracy of the proposed algorithm is 95%. These results demonstrate the efficacy and proficiency of the proposed hand gesture recognition algorithm in the MIMO radar framework.

## 5. Conclusions

In this work, an accurate and efficient radar-based hand gesture recognition system was proposed. The data were acquired from the radar equipped with 12 virtual antennas. It was then processed through range and velocity calculations utilizing a 2D-FFT. Moreover, the modified CA-CFAR with mirror filters and target detection algorithms was also adopted to enhance the robustness. The azimuth angle variable was determined via the MUSIC method. Then, an improved way for elevation angle computation was also proposed. Next, the defined parameters were determined adaptively, subjected to feature extractors, and

fed into the random forest system. The confusion matrix showed that the proposed gesture classification algorithm is robust, with an average of 95% overall accuracy across five folds. This result demonstrated the effectiveness of the proposed algorithm in accurately differentiating between eight movements. Additionally, the system performed well without requiring a large dataset or a complicated training process, highlighting its efficiency and applicability in gesture detection for healthcare and automatic driving applications. In our future work, we plan to delve into complex environments such as crowded spaces or dynamic settings where multiple individuals simultaneously make gestures. Moreover, the interference of the human body can be suppressed using the information of the relative depth and the amount of movement.

**Author Contributions:** Conceptualization, T.-J.T. and J.-J.D.; methodology, T.-J.T.; software, T.-J.T.; validation, T.-J.T.; formal analysis, T.-J.T. and J.-J.D.; investigation, T.-J.T.; resources, T.-J.T.; data curation, T.-J.T.; writing—original draft, T.-J.T.; writing—review and editing, T.-J.T. and J.-J.D.; visualization, T.-J.T.; supervision, J.-J.D.; project administration, J.-J.D.; funding acquisition, J.-J.D. All authors have read and agreed to the published version of the manuscript.

**Funding:** This research work was funded by National Science and Technology Council, Taiwan, grant number: MOST 110-2221-E-002-092-MY3.

**Data Availability Statement:** Data are contained within the article.

**Conflicts of Interest:** The authors declare no conflicts of interest.

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
