# Peer review of "Adaptive Trajectories’ Constant False Alarm Rate Mirror Filters and Elevation Angle Evaluation for Multiple-Input Multiple-Output Radar-Based Hand Gesture Recognition"

_electronics, doi:10.3390/electronics13040682_

Round 1
Reviewer 1 Report
Comments and Suggestions for Authors
The manuscript appears robust, and the results are commendable. The outline of the publication is clear and comprehensible. The manuscript could be recommended for publication in Electronics with a few corrections and additions. Below are comments for the authors to consider:
1. Authors should take note of MDPI Electronics Instructions for Authors (see References). "In the text, reference numbers should be placed in square brackets [ ], and positioned before the punctuation; for example, [1], [1–3], or [1,3]." "References must be numbered in the order of appearance in the text (including table captions and figure legends) and listed individually at the end of the manuscript."
2. The authors have utilized numerous abbreviations in the manuscript. I suggest inserting an Abbreviations section, or the authors could contemplate reducing their usage.
3. In the Results and Discussion section, it would be beneficial to explicitly state the advantages of the chosen methodology in this work and the created hand gesture recognition system compared to those described in existing publications.
4. It would be intriguing if the authors could briefly indicate potential more complex scenarios in the future where the interference suppression of the human body will be investigated.
Comments on the Quality of English Language
None
Author Response
Response to All Reviewers
In addition to the response to each reviewer, we also apply the following ways to improve the manuscript:
(1) Five more citations [1-5] are added.
(2) More description about the related work is given. Please refer to Section 2.
(3) We have invited someone to improve the use of English throughout this manuscript.
All of the revised parts are marked by yellow color. Please refer to the revised version of our manuscript. Many parts of the manuscript were rewritten and many new materials were added.
Response to Reviewer 1
>>> 1. Authors should take note of MDPI Electronics Instructions for Authors (see References). "In the text, reference numbers should be placed in square brackets [ ], and positioned before the punctuation; for example, [1], [1–3], or [1,3]." "References must be numbered in the order of appearance in the text (including table captions and figure legends) and listed individually at the end of the manuscript."
[Reply]: Thank you for the reminder. We have followed the suggestion and revised the format of the reference number under MDPI Electronics Instructions. We have followed the instruction to improve the style of presentation.
>>> 2. The authors have utilized numerous abbreviations in the manuscript. I suggest inserting an Abbreviations section, or the authors could contemplate reducing their usage.
[Reply]: As suggested, we have added a table to summarize all of the abbreviations used in this manuscript. Please refer to Table 2 at the end of Section 2.
>>> 3. In the Results and Discussion section, it would be beneficial to explicitly state the advantages of the chosen methodology in this work and the created hand gesture recognition system compared to those described in existing publications.
[Reply]: Thanks for your constructive suggestion. To clarify the advantage of the proposed method, we have added a paragraph in the second paragraph of Section 4.4 and rephrased the last two paragraphs in Section 4.4. Compared to deep learning-based algorithm, the proposed method can achieve excellent performance even if the amount of training data is limited. It requires much less training time to obtain an accurate classification model. It is suitable for the cases where the amount of training data is limited and quick analysis is required. We also compared the proposed algorithm to three conventional methods using the SVM, the KNN, and the NN. The proposed approach can achieve the highest accuracy with less training and testing time, highlighting its effectiveness in precisely identifying hand movements.
>>> 4. It would be intriguing if the authors could briefly indicate potential more complex scenarios in the future where the interference suppression of the human body will be investigated.
[Reply]: Thank you for the insightful suggestion. We recognize the importance of considering more complex scenarios of hand gesture recognition using the MIMO radar. As suggested, we added the description of the future work at the end of the conclusion section in the revised version. In our future work, we plan to delve into complex environments such as crowded spaces or dynamic settings where multiple individuals simultaneously make gestures. Moreover, the interference of human body can be suppressed by the information of the relative depth and the amount of movement.

Reviewer 2 Report
Comments and Suggestions for Authors
In this paper, a highly accurate hand gesture recognition system with low complexity is proposed. It applies the range-Doppler map (RDM) to generate range parameters adaptively and uses the elevation angle information. The idea is good written but it needs further improvements like:
1. improve the literature and compare with the proposed method to display the effectiveness.
2. make the contribution in points.
3. Prove that your proposed method is the best in comparison with others.
4. In this work, an accurate and efficient radar-based hand gesture recognition system 385 was proposed, How?
Comments on the Quality of English Languageminor editing
Author Response
Response to All Reviewers
In addition to the response to each reviewer, we also apply the following ways to improve the manuscript:
(1) Five more citations [1-5] are added.
(2) More description about the related work is given. Please refer to Section 2.
(3) We have invited someone to improve the use of English throughout this manuscript.
All of the revised parts are marked by yellow color. Please refer to the revised version of our manuscript. Many parts of the manuscript were rewritten and many new materials were added.
Response to Reviewer 2
>>> 1. Improve the literature and compare with the proposed method to display the effectiveness.
[Reply]: Thank you for the suggestion. We have added five new citations [1-5] of radar-based real-life scenario applications in Section 1. In Section 2, we have inserted Table 1 to compare the specifications of the related researches about hand gesture recognition in MIMO radar systems and summarized their adopted techniques. These methods rely on deep-learning-based approaches with large labeled training datasets and require complicated computational architecture, which limits their applicability in resource-constrained applications. We intend to develop a low-cost approach for MIMO radar-based hand gesture detection. Thus, the proposed algorithm is based on time-space feature extraction and the classifier of the random forest. Moreover, the result in Table 4 show that, compared to other conventional machine learning algorithms, the proposed method is more accurate and requires less training and testing time.
>>> 2. Make the contribution in points.
[Reply]: Thank you. As suggested, we added a paragraph to describe the main contribution of our work at the end of Section 1. The main contributions of our solution are as follows.
- This study uses adaptive signal processing techniques, including CA-CFAR, MUSIC, and elevation angle analysis, to extract the characteristics of the temporal trajectory of moving objects instead of using range-Doppler or range-angle images.
- The classification algorithm is based on the random forest method, which re-duces computing costs compared to the methods based on deep learning.
- Cooperating with a single-chip MIMO radar, our algorithm creates a low-complexity radar gesture recognition system suitable for healthcare and automatic driving applications.
>>> 3. Prove that your proposed method is the best in comparison with others.
[Reply]: Thank you for the suggestion. In Section 4.4, we have added several materials to compare the proposed method with other methods. Our approach the addresses classification problem in the case where the amount of training data is limited and quick analysis is required. Compared to deep learning-based algorithm, the proposed method can achieve excellent performance even if the amount of training data is limited. It requires much less training time to obtain an accurate classification model. We also compared the proposed algorithm to the methods using the SVM, the KNN, and the NN. The proposed approach can achieve the highest accuracy with less training and testing time, highlighting its effectiveness in precisely identifying hand movements.
>>> 4. In this work, an accurate and efficient radar-based hand gesture recognition system 385 was proposed, How?
[Reply]: Thank you for this comment. To clarify it, the time efficiency of the proposed algorithm has been described more clearly in Section 4. The SVM method meticulously selects a hyperplane, which slow down the process. The NN and the deep-learning based methods demand a large amount of data for effective parameter learning. In the case where the amount of training data is limited, the problem of overfitting may be led. Despite shorter testing times, the proposed approach achieves superior accuracy in both training and testing datasets, proving its effectiveness in hand gesture recognition. The experiment results show that the proposed method has a total average accuracy of 95%, with each gesture has the accuracy higher than 92%. Therefore, the proposed work is an accurate and efficient MIMO radar-based hand gesture recognition algorithm.

Reviewer 3 Report
Comments and Suggestions for Authors
The proposed idea to improve gesture recognition is exciting and appealing in this field. However, some questions must be addressed.
1) Table 1 and Table 2 presents a comparison with other methods. Did the authors test all these methods? If it is true, I consider that a more fair comparison can be made by comparing it with other papers.
2) The proposed algorithm strongly relies on the antenna array and the accuracy of the Texas Instruments evaluation boards. What will happen if other evaluation boards are used? Is the accuracy of the TI evaluation boards a dominant factor in achieving the 95% outstanding result? Is there any calibration or start-up method to quantify this effect?
3) Regarding Table 2. Please, add the power (energy) consumption of the proposed algorithm. This is very important because it limits the scenarios where this technology can be implemented. Also, this value can give important design constraints to the final user who may choose your algorithm or not. Everything in engineering is a trade-off.
Author Response
Response to All Reviewers
In addition to the response to each reviewer, we also apply the following ways to improve the manuscript:
(1) Five more citations [1-5] are added.
(2) More description about the related work is given. Please refer to Section 2.
(3) We have invited someone to improve the use of English throughout this manuscript.
All of the revised parts are marked by yellow color. Please refer to the revised version of our manuscript. Many parts of the manuscript were rewritten and many new materials were added.
Response to Reviewer 3
>>> 1) Table 1 and Table 2 presents a comparison with other methods. Did the authors test all these methods? If it is true, I consider that a more fair comparison can be made by comparing it with other papers.
[Reply]: We appreciate your comment. We have added several materials in Section 4 to explain that our approach focuses the classification problems where the amount of training data is limited and quick analysis is required. In contrast to the resource-intensive nature of modern deep learning methods, the proposed method can achieve excellent performance even if the amount of training data is limited. It requires much less training time to obtain an accurate classification model. We also compared the proposed algorithm to three conventional methods using the SVM, the KNN, and the NN and performed a comprehensive evaluation of all the methods. The MATLAB software was used to simulate all methods and all experiments were performed on identical datasets. A comparative analysis is conducted using the SVM with the one-vs-one strategy, the KNN algorithms with k set to five, and the NN method based on the feedforward technique with 50 epochs. The proposed approach can achieve the highest accuracy with less training and testing time, highlighting its effectiveness in precisely identifying hand movements.
>>> 2) The proposed algorithm strongly relies on the antenna array and the accuracy of the Texas Instruments evaluation boards. What will happen if other evaluation boards are used? Is the accuracy of the TI evaluation boards a dominant factor in achieving the 95% outstanding result? Is there any calibration or start-up method to quantify this effect?
[Reply]: Thank you for the comment. The proposed method is based on the IWR6843ISK radar model. If we use other radar devices, the dimension of the received data is different, which will affect the detection ability, the resolution of the parameters, and even impact the overall signal-to-noise ratio (SNR) of the radar. The proposed signal processing procedure is still applicable after proper calibration and modification for different devices. Please note that accuracy might be influenced by varying resolutions and data volumes. We have described the fact at the end of Section 4.3.
>>> 3) Regarding Table 2. Please, add the power (energy) consumption of the proposed algorithm. This is very important because it limits the scenarios where this technology can be implemented. Also, this value can give important design constraints to the final user who may choose your algorithm or not. Everything in engineering is a trade-off.
[Reply]: Thank you for this insightful suggestion. We have followed the suggestion to show the energy consumption of the proposed method and other methods in the last column of Table 4. All of the algorithms were implemented by Matlab, tested on the same laptop, and run 450 times. The results demonstrate that the energy consumption of the proposed algorithm is less than that of the NN method and the same as that of SVM in terms of the battery usage level, with the additional advantage of reducing computation time. Different algorithms can activate varying CPU computing units, which makes the relation between energy consumption and execution time not directly proportional. The computation time of the proposed algorithm is only 0,012 seconds and its energy consumption is only 0.00467 Wh.

Round 2
Reviewer 2 Report
Comments and Suggestions for Authors
All comments have been addressed
Comments on the Quality of English Languageminor editing